# Hypernatremia, Hyperlipemia and Hemorrhagic Enteritis in a Hypodipsic Dog with Corpus Callosum Dysplasia

**DOI:** 10.3390/ani15131996

**Published:** 2025-07-07

**Authors:** Pasquale Giannuzzi, Raffaella Perillo, Mariateresa Cafaro, Serena Paci, Clara Capogrosso, Michele Panarese, Debora Campanile

**Affiliations:** Ospedale Veterinario Atheryon, Via Grecia, 2, 76125 Trani, Italy; p.giannuzzi@hotmail.it (P.G.); raffaellaperilloneurologia@gmail.com (R.P.); dott.cafaromariateresa@gmail.com (M.C.); c.capogrosso89@gmail.com (C.C.); micpana@hotmail.it (M.P.); deboracampanile@gmail.com (D.C.)

**Keywords:** corpus callosum, hypernatremia, hyperlipemia, hypodipsia, hemorrhagic enteritis

## Abstract

We report a rare case of hypodipsia in a 7-month-old female Labrador Retriever with congenital corpus callosum dysplasia, presented for hemorrhagic enteritis. The hypodipsia-induced hypernatremic crisis was consistently associated with severe hyperlipemia, which resolved upon adequate water intake and sodium correction.

## 1. Introduction

Hypo-/adipsia associated with corpus callosal abnormalities is well documented in the veterinary literature [1,2,3,4,5,6,7,8,9,10], with hypernatremia-associated signs being the main reason for clinical presentation.

The prevalence of adipsia and hypernatremia in dogs with corpus callosal dysplasia is unknown; however, there is strong evidence suggesting a high association between this encephalic malformation and adipsia.

In rare cases, an increase in serum sodium appears to induce hyperlipemia, which is consistently associated with hypernatremia and resolves upon its correction.

While the association between hyperlipemia and hypernatremia is uncommon, it has been documented in a few pediatric cases [11,12,13,14,15,16], in a single canine case [17], and has also been investigated experimentally in rat models [18].

This case report describes a dog with hypodipsia associated with corpus callosum dysplasia, in which persistent hyperlipemia was observed during a hypernatremic crisis. The condition resolved following the restoration of adequate water intake and the subsequent correction in serum sodium levels. Additionally, this case highlights an atypical presentation of hypernatremia, characterized by hemorrhagic enteritis and suspected sepsis-associated liver dysfunction [19,20].

## 2. Case Presentation

A 7-month-old, intact female Labrador Retriever was admitted to “Atheryon Veterinary Hospital” in Trani (BT, Italy) due to apathy, generalized weakness, and diarrhea that had occurred over the past two days.

The owners reported that the dog “only sometimes” seemed interested in drinking water, prompting them to sporadically administer water forcibly by syringe.

The dog’s medical history included two previous episodes of apathy and weakness, occurring three and four months prior to the current evaluation. During these episodes, hematological tests consistently revealed hyperlipemia associated with hypernatremia and hyperchloremia. Furthermore, during the illness three months earlier, the patient had been hospitalized and received fluid therapy due to severe dehydration.

The persistent hyperlipemia remained of unknown origin. Clinical pathology findings excluded the most common causes of hyperlipemia, as urinary bile acids, the protein-to-creatinine (pu/cu) ratio, liver enzyme levels, and abdominal ultrasonography were all within normal limits. Hypothyroidism was not investigated at the time, as it was deemed highly unlikely given the patient’s age and clinical history.

On physical examination, the dog presented with severe depression, mild dehydration, and mild hyperthermia (39.3 °C). The pulse rate was 120 beats per minute, and the maximum blood pressure was 126 mmHg, with a capillary refill time of 2 s. Blood gas analysis performed upon admission revealed marked hypernatremia (195.8 mmol/L; reference interval [RI]: 144–150 mmol/L) and hyperchloremia (151 mmol/L; RI: 108–113 mmol/L). In the following hours, the dog developed melena.

Parvovirus and/or coronavirus infections were excluded using SNAP antigen tests.

Abdominal ultrasonography revealed a mildly thickened gastric wall with a slightly hyperechoic mucosal layer, moderate distension with corpusculated fluid content, and reduced motility. The small and large intestinal walls appeared normal but showed mild-to-moderate fluid content distension and overall reduced motility. Reactive jejunal lymphadenopathy was also observed. The kidneys, adrenal glands, and other abdominal organs were within normal limits.

The dog was hospitalized and treated with fluid therapy to gradually reduce sodium levels. Additional fluid volumes were administered to address dehydration and ongoing losses.

On the third day of hospitalization, a physical examination revealed jaundice. The patient exhibited an improved, though still depressed, mental status, with a body temperature of 39.3 °C, respiratory rate of 45 breaths per minute, and heart rate of 147 beats per minute. A complete blood count (CBC), standard biochemical profile, and urinary bile acids normalized to creatinine (UBA) were performed. Persistent hypernatremia (163 mEq/L; RI: 144–150) and hypercholesterolemia (471 mg/dL; RI: 179–355) were noted, while the triglyceride levels were within the reference interval (98 mg/dL; RI: 41–102).

Additional clinicopathological abnormalities included mild neutrophilia (8530/μL; RI: 3533–7284) with cytoplasmic toxicity; monocytosis (1070/μL; RI: 301–772); mild anemia (hemoglobin 11.7 g/dL; RI: 12.5–16.5); a marked elevation of C-reactive protein (7.49 mg/dL; RI: 0.01–0.4); hyperbilirubinemia (5.77 mg/dL; RI: 0.14–0.24); and increased ALP (586 IU/L; RI: 49–202), GGT (6.5 IU/L; RI: 1.4–5.3), and UBA (91.3 μmol/L; RI: 1–10.3). ALT and AST were within the reference interval.

The findings were deemed consistent with hepatic failure associated with sepsis, although a contributory role of concurrent hyperlipidemia and hypernatremia could not be ruled out.

In addition, while dehydration and hepatic dysfunction of presumed septic origin may have played a role in the development of hypernatremia associated with hyperlipidemia, the retrospective assessment of clinical and laboratory data preceding the onset of these abnormalities supports the hypothesis of a pre-existing chronic condition characterized by both hypernatremia and hyperlipidemia.

The owners declined consent for a blood culture. Empirical antibiotic therapy was initiated to monitor the patient’s clinical progression.

In the following days, the dog’s mental state improved significantly as its serum sodium levels decreased (Figure 1).

After 13 days of hospitalization, the dog was discharged in stable general condition. At discharge, a control biochemical profile revealed normalization of serum bilirubin (0.2 mg/dL; RI: 0.14–0.24), C-reactive protein (0.2 mg/dL; RI: 0.01–0.4), sodium (144 mEq/L; RI: 144–150), cholesterol (250 mg/dL; RI: 179–355), and triglycerides (73 mg/dL; RI: 41–102). A prescription was provided to ensure a daily water intake of at least 50 mL/kg, either mixed with food or administered forcibly with a syringe.

Based on the medical history, previous diagnostic tests, and findings during hospitalization, a diagnosis of chronic hypernatremia associated with hypodipsia, complicated by hemorrhagic enteritis and sepsis, was established. At this point in the diagnostic work-up, the transient hyperlipemia remained of unknown origin.

Three weeks later, the patient was re-evaluated, as the owners reported that the dog had not received the prescribed amount of water the previous days. On this occasion, a mild increase in serum sodium was observed, along with a moderate increase in cholesterol and triglycerides (Figure 2 and Figure 3). The dog was treated with fluid therapy for 24 h and subsequently discharged.

Considering the patient’s age, the suspicion of an encephalic malformation associated with dysregulation of the thirst mechanism was raised, and magnetic resonance imaging (MRI) of the brain was scheduled. Additionally, thyroid function tests (FT4 and TSH) were performed to rule out hypothyroidism as a potential cause of the hyperlipemia observed in previous blood tests. The thyroid function results were normal (TSH: 0.11 µg/mL; reference interval [RI]: 0.03–0.40; FT4: 29.7 pmol/L; RI: 12.8–47.3). MRI of the brain was performed with the patient in sternal recumbency using a small dog head coil (n°2-ESAOTE r) on a 0.25 T scanner (ESAOTE s.p.a., Genova, Italy). Standard FSE T2, SE T1, FLAIR, and 3D SST1 sequences were obtained in all three planes, with multiplanar reconstruction. Short TR sequences (T1 and 3D SST1) were repeated following intravenous gadolinium injection (0.2 mL/kg, Dotagraf Bayer).

The MRI findings revealed the absence of the rostrum, genu, and part of the trunk of the corpus callosum, with residual hypoplasia of the splenium and the caudal third of the body. The septum pellucidum, part of the fornix, and the ventral commissure were not identified. Symmetrical lateral ventriculomegaly with ventricular fusion and fusion of the cingulate gyri was observed, along with dilation of the pineal recess. The remaining cerebral parenchyma appeared normal in morphology and signal, and no pathological contrast medium uptake was detected (Figure 4).

A diagnosis of hypodipsia due to congenital corpus callosal dysplasia and holoprosencephaly (HPE) was established.

In the following months, the patient was managed by adding the daily fluid requirements to its food. Two additional hematological evaluations were performed monthly after the diagnosis, and the serum sodium, cholesterol, and triglyceride levels remained within the reference range (Figure 2 and Figure 3). At the final follow-up, six months after discharge, the patient was in normal general condition, with an unremarkable physical examination and the serum sodium concentration within the normal range (147 mEq/L; RI: 144–150).

## 3. Discussion

Hypo-/adipsia associated with corpus callosal abnormalities is well documented in the veterinary literature [1,2,3,4,5,6,7,8,9,10].

To the best of our knowledge, 35 cases of dogs with hypernatremia and hypo-/adipsia have been reported in the literature. Among these, 22 dogs were diagnosed with corpus callosum dysplasia. Other reported causes include encephalic lymphosarcoma (one case), primary adipsia (five cases), meningoencephalitis of unknown origin (one case), and hydrocephalus (one case), while five cases remained idiopathic [1,2,3,4,5,6,7,8,9,10,11,12,21].

The prevalence of adipsia and hypernatremia in dogs with corpus callosal dysplasia is unknown; however, there is strong evidence suggesting a high association between this encephalic malformation and adipsia. The most extensive case series of canine corpus callosum dysplasia, including 15 cases, reported that adipsia and hypernatremia were present in 12 patients (80%) [4].

In this dog with hypodipsia associated with corpus callosum dysplasia, persistent hyperlipemia was observed during the hypernatremic crisis. This condition resolved with the restoration of adequate water intake and the consequent correction of hypernatremia.

Although hyperlipemia associated with hypernatremia is rare, it has been reported in a limited number of cases in children [11,12,13,14,15,16] and one dog [17], as well as studied experimentally in rats [18].

A prospective case–control study in infants and children revealed that the majority of hypernatremic patients exhibited hypertriglyceridemia, which resolved once the serum sodium levels normalized [11]. Interestingly, the study suggested that this increase in triglyceride levels was not associated with acute hypernatremia but rather with the persistence of the hypernatremic state. Similarly, in a dog with a central nervous system lymphosarcoma and suspected hypothalamic osmoreceptor dysfunction, hypernatremia due to adipsia was associated with hyperlipemia. In this case, the hyperlipemia resolved following the correction of hypernatremia and recurred when the serum sodium levels increased [17]. Experimental studies in rats have also demonstrated an association between induced hypernatremia and hypertriglyceridemia [18].

While the association between hyperlipemia and hypernatremia is currently considered speculative, various mechanisms have been proposed that may suggest a potential direct or indirect influence of hypernatremia on lipid metabolism.

Damage to the ventromedial hypothalamus (VMH) region impairs fat mobilization by reducing the sympathetic tone or increasing the parasympathetic tone in adipose tissue. Adipose tissue, richly innervated by sympathetic fibers, relies on norepinephrine for lipolysis. Thus, VMH lesions disrupt this process [22,23].

Additionally, VMH lesions may influence lipid metabolism indirectly through insulin and glucagon secretion. Uninhibited ventrolateral hypothalamic function could lead to neurogenic stimulation of insulin secretion via the vagal nucleus, contributing to lipogenesis [24]. Glucagon excess in VMH lesions further promotes lipid synthesis [22].

However, in our case and others reported in the literature, hyperlipemia resolved with hypernatremia correction, making a concurrent hormonal or autonomic dysfunction less likely.

On the other hand, hyperosmolality caused by hypernatremia may directly enhance triglyceride (TG) formation in the liver. In vitro studies have shown that hypernatremia inhibits the conversion of fatty acids to ketone bodies in hepatocytes, favoring TG accumulation [25].

Lipoprotein lipase, an enzyme critical for clearing chylomicrons and very-low-density lipoproteins from circulation, is inhibited by hypernatremia. Crook et al. demonstrated that LPL activity is almost instantaneously inhibited by exposure to 500 mmol/L NaCl in vitro, with full reversibility upon dilution with a salt-free medium [26]. Chronic hypernatremia likely causes a gradual, sustained inhibition of LPL, contributing to hyperlipemia in vivo.

In our case, the patient exhibited hemorrhagic enteritis followed by hepatic insufficiency, likely related to an underlying septicemic state.

Interestingly, among 35 dogs with hypo-/adipsia and hypernatremia reviewed in the literature [1,2,3,4,5,6,7,8,9,10,21], 23% presented with gastrointestinal signs, including vomiting and/or diarrhea. These signs were generally mild and intermittent, often judged as clinically irrelevant.

The dog showed no evidence of exposure to toxic substances. Furthermore, parvovirus and coronavirus infections were ruled out. The most probable risk factors associated with the hemorrhagic enteritis complicated by sepsis were hyperlipemia and/or hypernatremia.

Studies evaluating the association between hypernatremia, hyperlipemia, and intestinal inflammation are scarce, as are documented cases reporting this rare complication. However, severe hypernatremia and hyperlipemia are known to cause plasma hypertonicity and hyperviscosity, respectively [27]. Cellular hyperosmotic stress occurs when extracellular osmolarity significantly exceeds intracellular osmolarity. This imbalance is particularly relevant in tissues such as the kidney, cornea, liver, gastrointestinal tract, intervertebral discs, and joints, which are naturally exposed to hyperosmotic fluids under physiological conditions [28].

Cells adapt to osmotic stress through mechanisms mediated by nuclear factor of activated T cell 5 (NF-AT5), also known as a tonicity-responsive enhancer-binding protein. This factor regulates gene transcription to restore osmotic equilibrium [29,30,31].

While the osmolarity of physiological fluids is tightly regulated, even slight deviations can have profound effects on cellular function. Hyperosmotic stress leads to cell shrinkage, disrupting DNA synthesis and repair, transcription, protein translation, mitochondrial function, and other homeostatic processes [29].

Most studies have focused on the effects of increased intestinal lumen osmolality. In experimental models, hyperosmotic stimuli such as alanine, mannitol, or NaCl in the colon induce inflammation, suggesting that any compound elevating colonic hypertonicity may have pathological consequences [32].

In human patients with inflammatory bowel disease (IBD), elevated fecal fluid osmolarity has been observed in the colon, with higher osmolarity correlated with disease severity [33,34,35,36]. While the impact of intestinal lumen hyperosmolarity is well documented, the effects of plasma hypertonicity on intestinal function remain unclear.

Although pancreatitis is the most common consequence of hyperlipemia, intestinal hemorrhages may also occur due to hyperviscosity syndrome associated with hyperlipemia.

A case of hypertriglyceridemia-induced enteritis has been reported in a 39-year-old man with uncontrolled diabetes mellitus. The authors hypothesized that lipemic serum may lead to acute colitis or that the hydrolysis of triglycerides into free fatty acids could inflame the gastrointestinal mucosa [37].

Hyperviscosity syndrome might provide another explanation for the intestinal bleeding observed in our patient. This syndrome involves abnormally elevated blood viscosity due to the pathological overproduction of blood components, including triglycerides [38].

The classic triad of hyperviscosity syndrome includes mucosal bleeding, visual abnormalities, and neurological symptoms. Blood viscosity is the highest in small venules, where the drag exerted by viscous fluids can tear the venule wall, particularly in areas lacking sufficient structural support. Consequently, bleeding is commonly observed in the nasal lining, gums, retina, gastrointestinal lumen, and brain surface [39].

## 4. Conclusions

This paper describes a case of hypernatremia associated with hyperlipemia in a hypodipsic dog diagnosed with corpus callosum dysplasia and holoprosencephaly. In this patient, the marked hyperlipemia resolved following the normalization in plasma sodium levels. To the best of our knowledge, this rare association has not been previously reported in dogs with corpus callosum dysplasia.

Although supported primarily by experimental and limited clinical data, the hypertriglyceridemia observed in chronic hypernatremia is likely multifactorial, with increased hepatic triglyceride secretion driven by hyperosmolality and the inhibition of lipoprotein lipase emerging as the most plausible mechanisms. These findings, while still preliminary, highlight the potential interplay between sodium homeostasis and lipid metabolism and underscore the importance of correcting hypernatremia in the management of associated metabolic disturbances.

Additionally, the development of severe hemorrhagic enteritis, complicated by sepsis and liver failure, represents an uncommon complication in cases of hypernatremia and/or hyperlipemia in dogs with encephalic malformations. In this case, intestinal hemorrhage may have been caused by the combined effects of plasma hypertonicity and hyperviscosity associated with hypernatremia and hyperlipemia.

Further studies are warranted to better understand the relationship between these factors and gastrointestinal complications. However, the observed pathophysiology aligns with existing evidence of the adverse effects of hyperosmotic and hyperviscous states, offering new insights into the clinical management of such cases.

## Figures and Tables

**Figure 1 animals-15-01996-f001:**
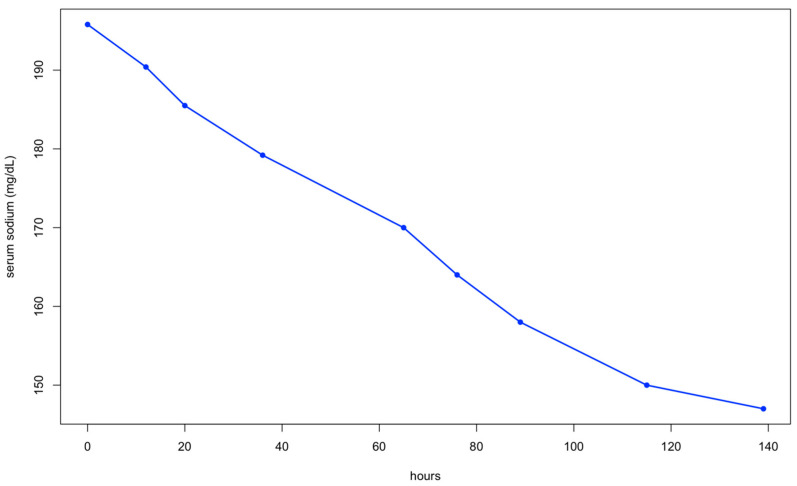
Serum sodium levels during the treatment of hypernatremia.

**Figure 2 animals-15-01996-f002:**
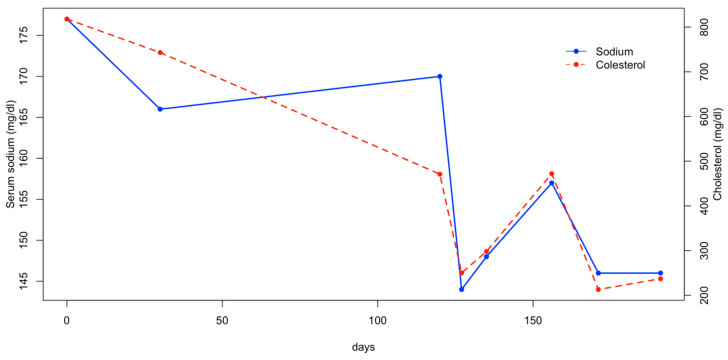
Serum sodium and cholesterol concentration.

**Figure 3 animals-15-01996-f003:**
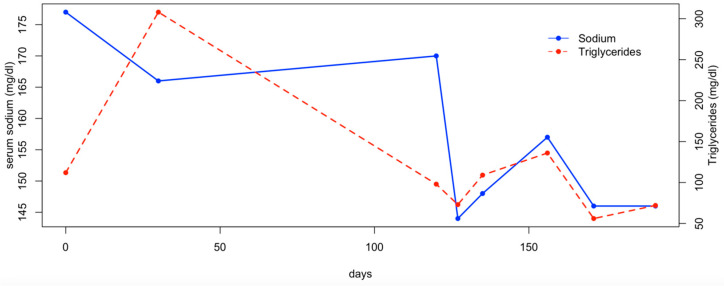
Serum sodium and triglycerides concentration.

**Figure 4 animals-15-01996-f004:**
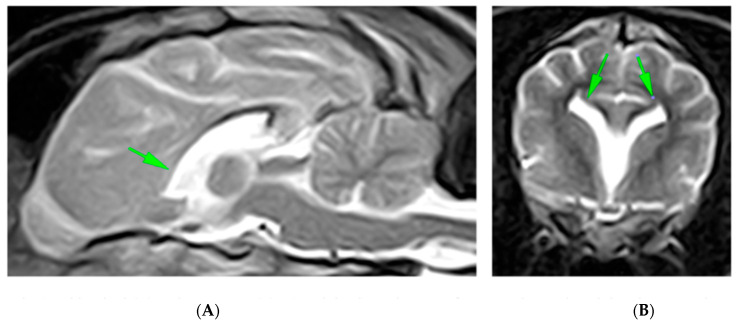
Midsagittal (**A**) and transverse (**B**) T2-weighted MR images of a 7-month-old crossbreed dog demonstrating fusion of the cingulate gyri, lateral ventricles, and ventral frontal lobes with associated loss of some normal midline structures. The rostral part of the corpus callosum is absent (arrowhead), and the caudal one is hypoplastic. On transverse MR images, the lateral ventricles have upturned, with pointed corners reminiscent of bat wings, as is typically seen with complete or focal aplasia of the CC (arrows).

## Data Availability

The original contributions presented in this study are included in the article. Further inquiries can be directed to the corresponding author.

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
