# Peer review of "Hypernatremia, Hyperlipemia and Hemorrhagic Enteritis in a Hypodipsic Dog with Corpus Callosum Dysplasia"

_animals, 2025, doi:10.3390/ani15131996_

Round 1

Reviewer 1 Report

Comments and Suggestions for Authors

Dear Editors of Animals and Authors,

This is a very interesting, well-written case report that documents the concurrent presence of hypernatremia and hyperlipidaemia in a hypodipsic Labrador Retriever puppy with corpus callosum dysplasia documented on MRI.
I was not aware of an association between hypernatremia and hyperlipidaemia, and I enjoyed learning about the proposed aetiology and pathophysiology in the discussion.

My major concerns are the limited information provided concerning laboratory tests and physical examination—for example, no information on the initial triglyceride blood levels or basic vital parameters—and the fact that the authors keep mentioning a “sepsis-associated liver dysfunction” without providing evidence for this diagnosis. The diagnosis of sepsis, which is considered to be a “systemic inflammatory response syndrome (SIRS) secondary to infection,” is already challenging in the best situations and relies on a combination of clinical signs, as reported below.

From TODAY’S VETERINARY PRACTICE | January/February 2015 | tvpjournal.com
From Sepsis chapter/Ettinger in VIN

In lines 81–86, the authors state:
“...the main clinicopathological findings included mild neutrophilia (8,530/μL; RI: 3,533–7,284) with evidence of neutrophilic cytoplasmic toxicity, increased C-reactive protein (7.49 mg/dL; RI: 0.01–0.4), elevated serum bilirubin (5.77 mg/dL; RI: 0.14–0.24), and increased UBA (91.3 μmol/L; RI: 1–10.3). These findings collectively suggested a diagnosis of suspected sepsis-associated liver dysfunction (20, 21), prompting the addition of metronidazole to the treatment regimen.”

I agree that there is liver dysfunction. However, they provide only scant information on the puppy’s physical examination, CBC, and biochemical profile. Unless the authors can provide strong evidence supporting their diagnosis of sepsis, I would recommend eliminating any reference to “sepsis-associated liver dysfunction,” or mentioning this only among other possible explanations. For example, we know that rats with induced hypernatremia exhibited significantly higher serum triglyceride levels and hepatic triglyceride content compared to controls, indicating that hypernatremia itself can lead to hyperlipemia and fatty liver. [Hypernatremia induces hyperlipemia and a fatty liver Hayek, Alberto et al. Metabolism - Clinical and Experimental, Volume 32, Issue 1]

The limitations of this case report, particularly the lack of information on triglyceride levels, do not preclude its publication but should be clearly acknowledged in the discussion.

I would also recommend adding arrows pointing to the described corpus callosum abnormalities in the MRI images in Figure 4.

Author Response

Dear reviewer 1,

we appreciated the valuable suggestions regarding the addition of insights into the physical and

hematological evaluation of the dog.

In this regard, we have added this information in the "Case presentation" section.

We, then, proceeded to add the appropriate graphic reference in the MRI images in Figure 4.

Reviewer 2 Report

Comments and Suggestions for Authors

In this case report, concurrent hypernatremia, hyperlipidemia, and hemorrhagic enteritis were observed in a hypodipsic dog with corpus callosum dysplasia. Improvement in these parameters was achieved with fluid therapy, and the authors suggest that these findings may be associated with systemic effects stemming from hypodipsia. They particularly emphasize the parallel changes between hypernatremia and hyperlipidemia, proposing a potential link between these two parameters.

Some findings presented in the case do support this hypothesis. Notably, the tables provided by the authors show that sodium and cholesterol levels increased and decreased in a similar pattern over time, and triglyceride levels also followed a similar trend. This correlation is noteworthy and is supported by some reports in the literature.

However, despite these findings, the data provided in the case are insufficient to establish a causal relationship, and the conclusions drawn are presented in a tone that is overly definitive. The following limitations and shortcomings should be considered:

  1. Although the values appear to change in parallel, no mechanistic data are provided to determine whether there is a direct causal relationship underlying this co-occurrence. The simultaneous increase is more likely due to a shared physiological origin—particularly hemoconcentration caused by sepsis and dehydration. Under such conditions, both lipid and sodium levels may increase in parallel; however, concluding that one directly affects the other exceeds the limits of the presented data.
  2. Findings observed in the early stage of the case—such as melena, hyperthermia, neutrophilia, elevated bilirubin, and bile acid levels—strongly suggest sepsis. Notably, after initiating metronidazole treatment, not only sodium and lipid levels but also bilirubin and clinical status improved. This implies that systemic inflammation may affect both water-electrolyte balance and lipid metabolism. However, the possibility of changes in sodium and lipid metabolism due to sepsis is not adequately addressed in the discussion. Attributing the changes solely to fluid therapy overlooks the potential impact of antibiotic treatment. Furthermore, no information is provided regarding any microbiological cultures performed to identify a septic agent.
  3. Although the authors refer to the roles of hormones like insulin and glucagon in lipid metabolism from the literature, insulin or glucagon levels specific to this case were not measured. Similarly, enzymatic data such as LPL (lipoprotein lipase) activity, which could directly reflect metabolic processes, are absent. These omissions render the metabolic interpretations speculative.
  4. While hyperbilirubinemia and elevated bile acid levels suggest liver dysfunction, associated liver enzyme levels (ALT, AST, ALP, GGT) are not presented. Considering the central role of the liver in lipid metabolism, it is not possible to explain the alterations in lipid levels without these parameters.
  5. In the information provided about the animal’s follow-up and treatment in recent months, it is only stated that blood parameters normalized with fluid therapy. However, there are no data on whether infection was ruled out during this period using CRP, WBC, or liver enzyme levels. Additionally, critical information such as the animal’s overall condition, survival status, and duration of follow-up is lacking.

This case presents some intriguing observational findings and offers a potentially valuable insight into the relationship between hypodipsia, hypernatremia, and hyperlipidemia. However:

  • The conclusions are presented with a level of certainty that exceeds what the available data support.
  • Alternative explanations—such as the direct effects of sepsis and dehydration on sodium and lipid metabolism and their potential to improve independently following resolution—were overlooked.
  • Hormonal and enzymatic data that would support metabolic interpretations are missing from each treatment and control stage.
  • The clinical and laboratory aspects of the follow-up process are insufficiently detailed.

Author Response

First of all, we are very grateful to the reviewers for their valuable comments, which will

undoubtedly improve the reporting of this paper. Based on the suggestions provided, we

have revised several sections of the manuscript. In particular, the changes addressing the

comments from Reviewer 1 and points 1 and 2 from Reviewer 2 are incorporated in the text

between lines 87 and 105.

Moreover, and also with the pleasure of sharing our assessment of the case, we would like to

emphasize that in this case, the suspected associa/on between hypernatremia and

hyperlipidemia (hypercholesterolemia and/or hypertriglyceridemia) was not based solely on

the evalua/on of the pa/ent at the /me of hemorrhagic diarrhea and suspected sepsis, but

rather on the en/re temporal profile (before and a=er the complica/on), which we consider to

be well documented. Two months prior, this dog had undergone a complete hematological

and biochemical evalua/on (37 parameters assessed), including serum protein

electrophoresis. The only abnormali/es detected at that /me were marked

hypercholesterolemia (818 mg/dL; RI: 179–355), mild hypertriglyceridemia (112 mg/dL),

moderate hypernatremia (156 mEq/L; RI: 144–150), and a mild increase in ALP, the laUer

deemed age-related. At that /me, there was no clinical or laboratory evidence of

inflamma/on or hepa/c dysfunc/on. We believe the ini/al interpreta/on may have been

misleading, as it is well known that indirect sodium measurement can underes/mate serum

sodium concentra/on in the context of hyperlipidemia. One month before our assessment,

the pa/ent had undergone an even more extensive hematological check-up, which included a

coagula/on profile and urinary bile acid quan/fica/on. Again, at that /me, the dog showed no

clinical or clinicopathological evidence sugges/ve of sepsis. Urinary bile acids were within the

reference interval (7.0 µmol/L; RI: 1–10), yet marked hyperlipidemia was observed

(cholesterol 742 mg/dL; triglycerides 308 mg/dL) in associa/on with hypernatremia (168

mEq/L).

Taken together, these data—along with the clinical course a=er discharge, during which

cholesterol and triglyceride levels normalized in parallel with normaliza/on of serum

sodium—support the hypothesis of an associa/on between hypernatremia and

hyperlipidemia, similar to what has been reported in pediatric cases and in the dog we

previously described in the manuscript.

Further suppor/ng this hypothesis, a noteworthy event occurred three weeks post-discharge:

the owners failed to administer water forcibly to the dog, resul/ng in recurrent

hypernatremia, which was again accompanied by a moderate increase in cholesterol andtriglyceride levels. This laUer episode strengthens the likelihood that hyperlipidemia is

consistently associated with hypernatremic episodes in this pa/ent.

1. The hypothesis of a possible contribution from hypothalamic hormonal

dysfunction was considered unlikely by us, as it would not account for the

resolution of hyperlipidemia achieved through correction of the patient’s

hydration status. On the other hand, we have already addressed this concept in

another section of the manuscript: '...However, in our case and others reported in

the literature, hyperlipemia resolved with hypernatremia correction, making a

concurrent hormonal or autonomic dysfunction less likely...'(line 192-194).

2. We were not able to assess lipoprotein lipase (LPL) ac/vity in this case; therefore,

the proposed hypothesis remains specula/ve and aligns with what has already

been reported in the literature for similar cases. As emphasized in the Discussion

sec/on, the correla/on between hypernatremia and hyperlipidemia is supported

by in vitro and animal studies demonstra/ng the following: (1) hypernatremia can

inhibit lipoprotein lipase ac/vity, as reported in a pediatric case of

hypertriglyceridemia associated with hypernatremia due to a hypothalamic tumor

(Crook M, Robinson R, Swaminathan R. Hypertriglyceridemia in a Child with

Hypernatremia due to a Hypothalamic Tumor); (2) LPL is inhibited by 500 mmol/L

NaCl in vitro, and this inhibi/on is fully reversible (Fielding CJ. Inac7va7on of

Lipoprotein Lipase in Buffered Saline Solu7ons); and (3) hypertriglyceridemia

developed in rats when mean serum sodium concentra/ons exceeded 159 mmol/L

(Hayek et al. Hypernatremia Induces Hyperlipidemia and Fa@y Liver).

3. the changes addressing this comment are incorporated in the text between

lines 95-96

4. The requested data have been added. We considered the dog to be in remission

with respect to hepa/c failure and the inflammatory state, as both CRP (added in

line 112) and bilirubin levels were within the normal range

Finally, in order to beUer align the level of certainty in our conclusions and discussion

with the available data, we have revised several parts of the manuscript: Lines 187-

189; Lines 268 – 274.

If the reviewers consider that further clarifica/on is needed, we would be happy to revise the

relevant sec/on of the manuscript accordingly.

Round 2

Reviewer 1 Report

Comments and Suggestions for Authors

Dear Editors of Animals and Authors,
Thank you for addressing my previous concerns.
I just have a couple of additional minor notes:
Please spell out Ventromedial Hypothalamus in full before introducing the abbreviation (VMH).
I suggest adjusting the arrows as shown here to more accurately indicate the position of the corpus callosum.
The corpus callosum is typically located superior to the "bat wing" structures; therefore, the arrows should be adjusted to clearly indicate the pointed corners, as demonstrated below. Additionally, I recommend removing non-neural structures such as the nose, neck, pharynx, and masticatory muscles from the figure, as they do not contribute to the neural focus of the illustration and may be visually distracting.

Author Response

Comment Reviewer 1

Dear Editors of Animals and Authors,

Thank you for addressing my previous concerns.

I just have a couple of additional minor notes:

Please spell out Ventromedial Hypothalamus in full before introducing the abbreviation (VMH).

I suggest adjusting the arrows as shown here to more accurately indicate the position of the corpus

callosum.

The corpus callosum is typically located superior to the "bat wing" structures; therefore, the arrows should

be adjusted to clearly indicate the pointed corners, as demonstrated below. Additionally, I recommend

removing non-neural structures such as the nose, neck, pharynx, and masticatory muscles from the figure,

as they do not contribute to the neural focus of the illustration and may be visually distracting.

Answer

Thank you for welcoming the improvements. We have also changed the abbreviation and image as

requested.

Reviewer 2 Report

Comments and Suggestions for Authors

The authors have provided the necessary arrangements, additions and explanations in line with my suggestions and the data they have. The article has made significant progress, especially in terms of detailing clinical information in various time periods, presenting some missing parameters, adding follow-up information and editing ambitious sentences. Although there are deficiencies due to some data that are not available, I believe that the study can be a focus of interest for the readership. For this reason, I recommend that the study be accepted.

Author Response

Comment Reviewer 2

The authors have provided the necessary arrangements, additions and explanations in line with my

suggestions and the data they have. The article has made significant progress, especially in terms of

detailing clinical information in various time periods, presenting some missing parameters, adding follow-

up information and editing ambitious sentences. Although there are deficiencies due to some data that are

not available, I believe that the study can be a focus of interest for the readership. For this reason, I

recommend that the study be accepted.

Answer

Thank you for appreciating the improvements and details added to our case report. We have done our

best to make it more relevant to the questions posed to us.